# Analysis of the Effects of Highway Geometric Design Features on the Frequency of Truck-Involved Rear-End Crashes Using the Random Effect Zero-Inflated Negative Binomial Regression Model

Thanapong Champahom [1], Chamroeun Se [2], Sajjakaj Jomnonkwao [2,*], Rattanaporn Kasemsri [3] and Vatanavongs Ratanavaraha [2]

1    Department of Management, Faculty of Business Administration, Rajamangala University of Technology Isan, Nakhon Ratchasima 30000, Thailand; thanapong.ch@rmuti.ac.th
2    School of Transportation Engineering, Institute of Engineering, Suranaree University of Technology, Nakhon Ratchasima 30000, Thailand; chamroeun.s@g.sut.ac.th (C.S.); vatanavongs@g.sut.ac.th (V.R.)
3    School of Civil Engineering, Institute of Engineering, Suranaree University of Technology, Nakhon Ratchasima 30000, Thailand; kasemsri@sut.ac.th
*    Correspondence: sajjakaj@g.sut.ac.th

**Abstract:** Statistical data indicate that trucks are more prone to rear-end crashes, making this an area of concern. The objective of this study is to create a model that analyzes the factors influencing the frequency of rear-end crashes involving trucks (TIRC). To achieve this, researchers identified the most appropriate model as Spatial Zero-Inflated Negative Binomial Regression (SZINB). This model takes into account spatial correlation, which plays a significant role in the occurrences of TIRC on different road segments supervised by each highway ward. The estimation of parameters in the SZINB model has led to key findings that shed light on the factors contributing to a higher likelihood of TIRC. These findings include the increased probability of TIRC on curved roads compared to straight ones, roads that feature open middle islands, six lanes per direction, a slope, right-of-way shoulder width, pavement type, lane width, and a post speed limit. Based on these key findings, this study developed policy recommendations and sample measures aimed at reducing the frequency of TIRC. Implementing measures such as improving the road design on curved sections, optimizing middle islands, and enhancing traffic management on wider roads can help mitigate the risk of crashes involving trucks.

**Keywords:** crash modification factor; Thailand; developing country; spatial correlation; curve; slope

## 1. Introduction

### 1.1. Research Background

Owing to the long-standing emphasis on the development of road networks within the transportation infrastructure, truck transportation has assumed a pivotal and indispensable role in Thailand. The nation's logistics system predominantly relies on road transport, contributing to the steady growth of trucking companies. As of 2021, the cost of freight transport by road was projected to account for a substantial 68.88% [1]. While the Thai government is making efforts to enhance railway infrastructure with the aim of reducing logistics costs, the implementation of such improvements still requires considerable time. As a result, trucks are poised to retain their significance and continue playing a crucial role in the transportation sector for the foreseeable future.

Despite the numerous advantages of trucking, such as its speed and door-to-door accessibility, it inevitably carries a significant disadvantage as follows: a high risk of crashes. These crashes can be caused by various factors, including drivers falling asleep or navigating through challenging terrains, such as steep slopes [2,3]. Analyzing truck crash data from

Thailand between 2018 and 2020, classified by collision type and severity levels (as shown in Table 1), it becomes apparent that the most common type of collision is a single collision, closely followed by rear-end collisions. However, when considering the severity of these crashes, head-on collisions present the highest chance of fatalities, accounting for 46.83%, followed by rear-end collisions with a fatal crash probability of 24.36%. Consequently, in light of these findings, policy recommendations aimed at addressing road safety issues should prioritize rear-end collisions due to their higher frequency and severity.

**Table 1.** Truck involvement collision type and crash severity.

| Collision Type/Crash Severity | Fatal Crash | | Severe Crash | | Minor Injury Crash | | PDO | | Total |
|---|---|---|---|---|---|---|---|---|---|
| Pedestrian collision | 70 | 14.55% | 33 | 6.86% | 104 | 21.62% | 274 | 56.96% | 481 |
| Rear-end collision | 930 | 24.36% | 517 | 13.54% | 1139 | 29.84% | 1231 | 32.25% | 3817 |
| Sideswipe collision | 272 | 19.93% | 125 | 9.16% | 412 | 30.18% | 556 | 40.73% | 1365 |
| Single collision | 308 | 4.59% | 309 | 4.60% | 1697 | 25.26% | 4403 | 65.55% | 6717 |
| Head on collision | 236 | 46.83% | 91 | 18.06% | 118 | 23.41% | 59 | 11.71% | 504 |

Note: A "fatal crash" is characterized as an incident in which at least one individual succumbs to their injuries. On the other hand, a "severe crash" refers to a crash in which at least one person sustains serious injuries but there are no fatalities. A "minor injury crash" denotes a crash involving at least one minor injury without any instances of serious injury or death. Lastly, "PDO" (Property Damage Only) signifies a crash with no reports of minor injuries, serious injuries, or fatalities.

Crashes are always the outcome of a combination of variables regarding the following three factors: driver(s), vehicle(s), and the road environment. Baikejuli, et al. [4]. emphasized that a significant portion of crashes can be attributed to human drivers. However, it is essential to recognize the substantial impact of road conditions on driver behavior. For instance, excessively long, straight roads can induce drowsiness in drivers, especially those operating trucks for extended periods, as highlighted by Song, et al. [3]. Such fatigue-related factors increase the risk of sleep-related crashes. In the absence of safety measures like shoulder rubble strips, microsleep warning signs, and other road features, the likelihood of crashes tends to rise significantly, as observed by Al-Bdairi and Hernandez [5]. Addressing road factors, specifically concerning rear-end collisions, it is revealed that areas with steep inclines pose a similar risk of rear-end collisions as roadside locations. This heightened risk is attributed to the extended steep slopes that may cause truck brakes to malfunction due to excessive usage, leading to insufficient stopping power. Apronti, et al. [6]. supports this notion, confirming the possibility of high rear-end collision rates.

While several developed countries have made significant strides at reducing road traffic fatalities in recent years, progress remains highly variable worldwide. The risk of road traffic fatalities is considerably higher in low- and middle-income countries, with an average rate of 27.5 per 100,000 population, compared to high-income countries, which have an average rate of 8.3 per 100,000 population [7]. Thailand, classified as a middle-income and developing nation, bears substantial economic and emotional burdens due to road accidents, with a death rate of 32.8 per 100,000 population [7]. According to the Department of Highways (DOH) report in 2004, the cumulative costs attributed to traffic accidents in Thailand amounted to THB 153,755 million (equivalent to approximately USD 3460 million) [8]. Furthermore, considering the data from 2017, the total productivity loss solely due to road traffic accidents reached approximately THB 121 billion. This figure includes THB 45 billion from fatalities, THB 7 billion from disabilities, THB 67.5 billion from serious injuries, and THB 1.5 billion from minor injuries. Remarkably, these losses account for nearly 0.8% of the country's gross domestic product [9]. These roadway accidents are primarily attributed to widespread speeding practices that frequently exceed legal speed limits [10]. Adding to this concern is the extensive presence of motorcycles on the road, sharing the same space with larger and faster vehicles, such as cars and trucks [11]. Together, these factors contribute to an increased frequency of accidents. Given the nature of the construction and road network development in Thailand, This country is divided into 77 provinces, each overseen by 18 departments of highways (DOH). These DOH

branches are responsible for managing road infrastructure and development within their respective provinces. However, in some instances, larger provinces may have more than one DOH branch (Figure 1a). While the DOH's work adheres to the same standards, there are variations in visual characteristics and policies across different districts, as well as differences in land use. These variations can potentially impact road design and may contribute to rear-end collision crashes involving trucks. Figure 1b illustrates that the frequency of crashes differs across provinces and regions under DOH supervision, with higher occurrences in the central and eastern regions, such as Bangkok, Pathum Thani, and Chonburi. This can be attributed to the concentration of industrial activity in these areas, leading to a significant volume of truck traffic. Numerous studies have conducted spatial analyses using random effect models to examine various aspects of crashes. For example, Lee, et al. [12] explored the proportion of crashes for each type of vehicle at the traffic analysis zone level. Wen, et al. [13] focused on crash frequency and severity, analyzing the expected crash frequency on freeway segments. Osama and Sayed [14] investigated spatial correlations within traffic analysis zones in the City of Vancouver. The results of their research indicate that spatial correlations are influenced by the inclusion of explanatory variables in the model.

### 1.2. Research Gaps and Objectives

Most studies pertaining to truck crashes predominantly focus on the magnitude of the injury scale. For instance, Xu, et al. [15] conducted research on road-related violence, wherein they measured it as the ratio of fatal collisions and injured collisions combined, divided by the total number of collisions in the specific road segment. Wen, et al. [2] delved into examining the severity of injury levels, specifically analyzing the impact of curves and slopes on multi-vehicle truck crashes. In another investigation, Wang, et al. [16] scrutinized the severity of injury between crashes involving trucks and those not involving trucks. The simulation aspect has been taken into account, along with spatiotemporal instability. Duvvuri, et al. [17] studied the degree of injury severity as well as the introduction of novel factors such as land use and demographic variables.

When investigating rear-end collisions, specifically concerning crash frequency, researchers have conducted notable studies that shed light on this phenomenon. For instance, Champahom, et al. [18] compared the occurrences of rear-end collisions by examining spatial variations. Meanwhile, Lao, et al. [19] analyzed the risk of such collisions utilizing a generalized nonlinear model (GNM). Notably, both of these studies employed road imagery factors to establish correlations with the frequency of rear-end collisions.

Moreover, researchers have delved into the examination of rear-end collision frequencies within specific regions. Ref. [20] focused on work zones, whereas [21] explored signalized intersections. Additionally, for a more specialized perspective on the frequency of truck-involved rear-end crashes (TIRC), Yuan, et al. [22] conducted a comprehensive investigation. Notably, their study revealed that factors such as the rider's age and the type of vehicle in front significantly impact the severity of injuries in TIRC cases.

Despite not investigating new variables or introducing novel computational methods, this study is groundbreaking in its own right. Prior to this research, no study examined the frequency of TIRC. Therefore, the primary objective of this study is to fill this knowledge gap and enhance an understanding of rear-end collision prevention. To achieve this objective, researchers aimed to construct a model that could establish a connection between highway geometric factors and the occurrence of TIRC. Significantly, this model also accounts for variations across different Department of Highway branch (DOHB) control areas. The study's scope was confined to Thailand's highways, which boast the highest proportion of truck traffic due to their vital links between major cities in the country. To gain deeper insights into the influence of highway geometric design factors on TIRC frequency, researchers employed crash modification Factors, which capture the variations in variables affecting crash frequency [23]. In order to specify the methodology for conducting this

study, the research question we aim to address is as follows: how do highway geometric design features impact the frequency of rear-end collisions involving trucks?

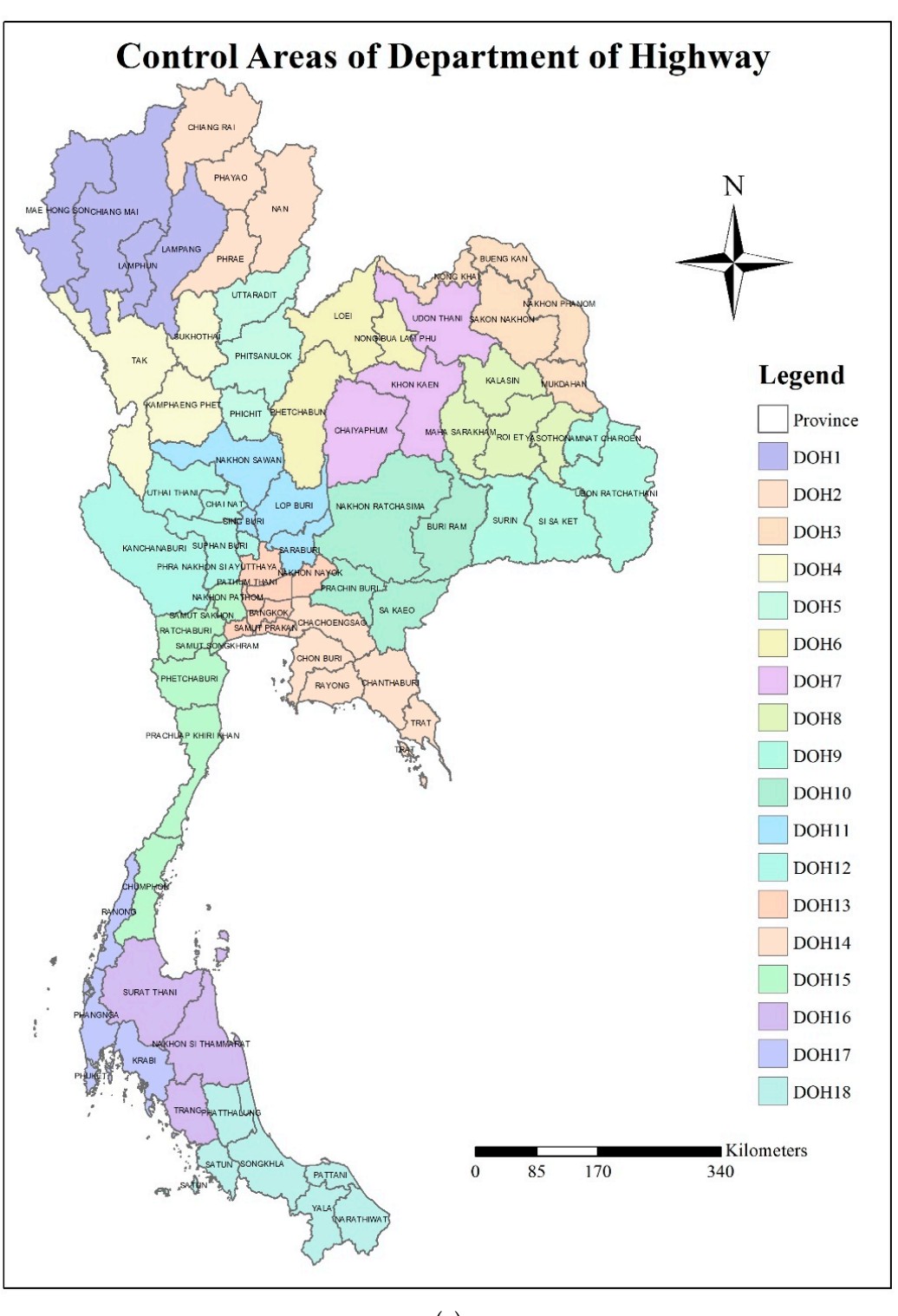

(**a**)

**Figure 1.** *Cont.*

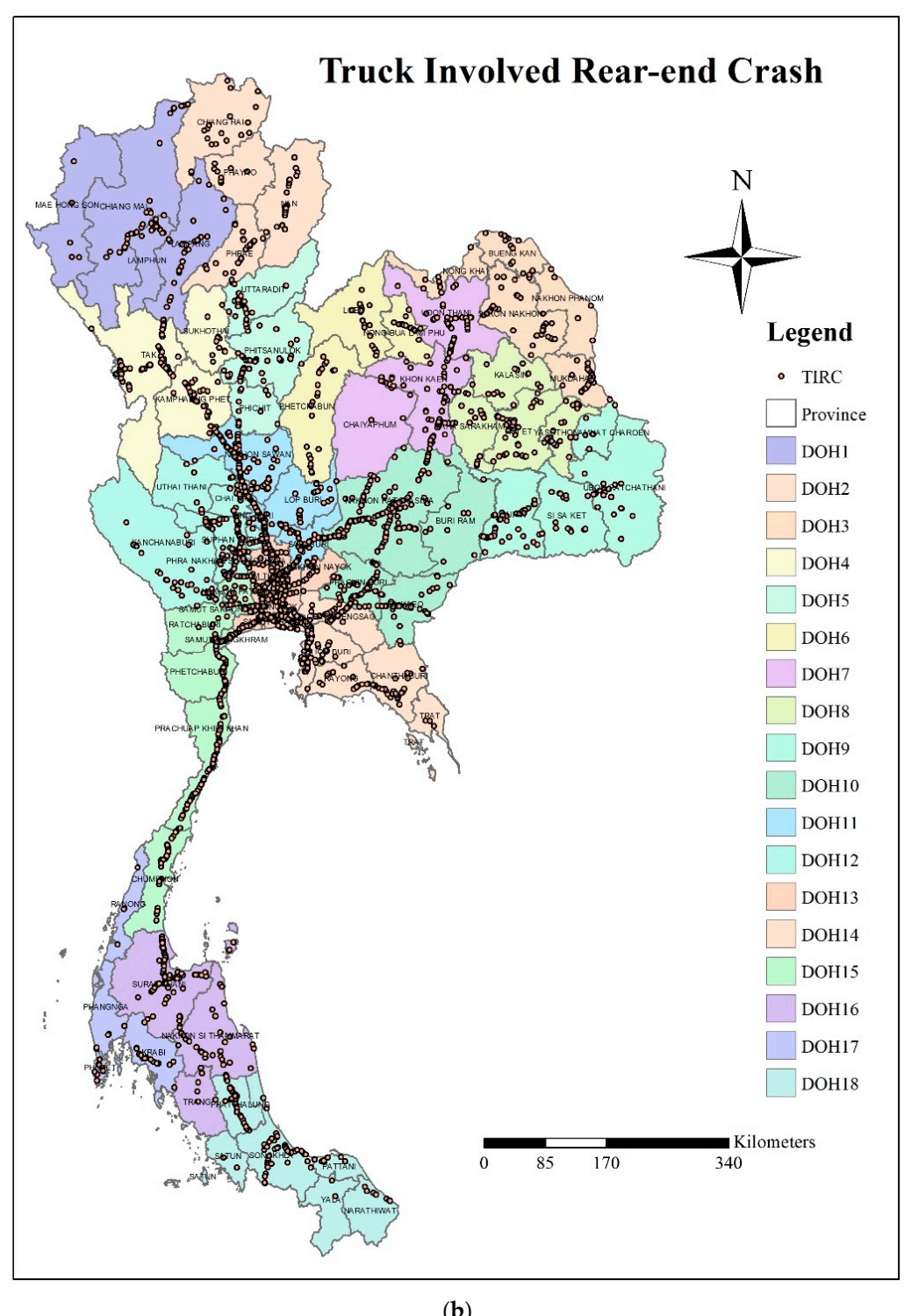

(**b**)

**Figure 1.** (**a**) Control areas of Department of Highways Office (DOH) and (**b**) Truck-Involved Rear-end Crash Points. The colors within the diagram denote the geographical jurisdictions of the Department of Highway offices. To illustrate, the symbol 'DOH1' signifies Department of Highway Offices Number 1, encompassing their operational purview over four provinces situated in the northern region of the country, namely Mae Hong Son, Chiang Mai, Lamphun, and Lampang.

The substantial contribution of this study lies in the conversion of knowledge derived from this model's outcomes into policy recommendations. These recommendations inform the development of physical features that effectively minimize the likelihood of rear-end

collisions with trucks. By doing so, this research has the potential to significantly enhance road safety and reduce crashes involving trucks.

## 2. Highway Geometric Design Features on the Frequency of Truck-Involved Crashes

Based on previous research, it has been established that numerous factors possess the potential to influence the frequency of truck-involved crashes. Let us first delve into the fundamental factors that several studies have identified as impacting crash frequency. The foremost factor is the average annual daily traffic (AADT), which has been consistently correlated with the frequency of crashes [24,25]. As traffic volume increases, the likelihood of collisions on the road also rises. Champahom, et al. [11] elucidated that heightened car volume leads to reduced headway between vehicles, resulting in shorter braking distances and, consequently, an increased risk of rear-end collisions.

Another crucial factor contributing to this phenomenon is the proportion of trucks on the road. It is evident that a higher proportion of trucks leads to an increased likelihood of rear-end collisions involving trucks as well [26]. Additionally, the length of each road segment has been found to exert a positive effect on collision frequency [19].

For the group of variables concerning the physical characteristics of the road, one significant factor is the number of traffic lanes. The frequency of rear-end collisions is indirectly related to the number of traffic lanes. According to Wen, et al. [27], an increase in traffic lanes aims to facilitate faster car flow or higher speeds (increased AADT) when braking suddenly, which, in turn, raises the likelihood of collisions. This effect is particularly pronounced for trucks with longer braking distances compared to light trucks, further increasing the chances of rear-end collisions. Another influential factor is the posted speed limit, which shows a clear correlation with speed-related collisions. Wen, et al. [27] suggest that reducing the speed limit by law could potentially decrease the frequency of collisions. The type of road surface, such as asphalt concrete and concrete, plays a role in the frequency of rear-end collisions due to differences in friction. Concrete surfaces generally offer more friction than asphalt concrete. However, there have been varying findings from previous research. While Champahom, et al. [18], Feknssa, et al. [28] reported insignificant differences between road surface types, a study by Iranitalab, et al. [29] found that concrete surfaces are less likely to be involved in crashes. The width of the traffic lane is another variable of interest. In Thailand, most roads adhere to standards with lane widths ranging from 3.1 to 3.5 m, but some segments use wider lanes specifically designed for trucks. Agbelie [30] observed that an increase in lane width leads to a positive association with the frequency of collisions, meaning wider lanes are linked to higher chances of crashes. The presence of footpaths also affects collision probability as it indicates the presence of pedestrians or significant activities in that area, leading to increased traffic lights and heavier traffic volume. Dong, et al. [31], Dong, et al. [32] demonstrated that more traffic control devices contribute to a higher chance of rear-end collisions. Shoulder width, on the other hand, appears to have little to no significant effect on the frequency of crashes, as suggested by Dong, et al. [33]. Right-of-way is an essential factor in defining road boundaries and reflects connections between major cities, emphasizing traffic flow mobility. Roads with a high number of traffic lanes are often built in such areas [34]. Regarding median width, Champahom, et al. [11] found a significant association between central island width and the likelihood of crashes, with wider medians having a higher chance of collisions. In terms of visual characteristics, straight roads are generally less prone to rear-end collisions compared to curved roads. Islam, et al. [35] highlighted that curved roads necessitate a reduction in speed, particularly in cases of high traffic volume, increasing the risk of rear-end collisions. Slope variation is another factor influencing the frequency of rear-end collisions, especially for trucks, due to brake-related issues when repeatedly applying the brakes [36]. Finally, the presence of median openings also contributes to the likelihood of rear-end collisions, as observed by Champahom, et al. [37]. Open center islands require vehicles to reduce their speed, and if drivers are not careful or the vehicle ahead does not provide a proper turn signal, rear-end collisions may occur.

## 3. Methods Section

### 3.1. Data Collection

Data regarding crashes on Thailand's highways are meticulously gathered and recorded in the highway crash information management systems (HAIMS), which are maintained by the Department of Highways. The acquisition of crash cases is carried out by designated officials from various highway districts across different provinces, who conduct regular visits to crash sites. During these visits, they collect a multitude of pertinent factors, such as the physical characteristics of the crash site, precise location details (e.g., highway no. 2 km. 20 + 200), and the slope of the road, among others.

For this particular study, a specific type of crash, namely rear-end collisions involving shared trucks, was chosen for analysis. The dataset utilized for this study encompasses crashes that occurred between 2011 and 2018. Notably, data from the years 2019 to 2021 were not included in the analysis due to the disruptive impact of the COVID-19 outbreak. This unprecedented situation led to abnormal behavior patterns and altered traffic volume, especially with regard to the amount of cargo being transported, thereby directly affecting the number of trucks involved in crashes [38].

Traffic volume and road geometry data are gathered within the traffic information management system (TIMS) administrated by the Department of Highways. This comprehensive system provides intricate information about each road, including its road number, initial and final kilometer points, road length, traffic surface type, lane width, and road shoulder width, among other details.

Consequently, the average annual daily traffic (AADT) and the Percentage of Trucks (truck percentage) were correlated with the highway number and control section. The resulting traffic volume data were subsequently stored in the TIMS database. Similarly, matching between crash points and road segments was achieved by aligning the road number and column kilometer.

Data pertaining to truck-involved rear-end crashes (TIRC) are displayed in Table 2. This table encompasses 16,936 segments, each with an average distance of 3.082 km (standard deviation = 5.023). The number of crashes in these segments was recorded as 0.291 (standard deviation = 2.071). The average daily traffic volume was found as 14,473.9 vehicles, with trucks accounting for 16.328 vehicles per day. Regarding road geometry, the majority of traffic lanes consist of 2 lanes and 4 lanes, constituting 57.32% and 32.23%, respectively. It is important to note that the figures presented for curve, slope, and median opening numbers are only applicable to segments where at least 1 crash occurred. Segments without crashes but with curves were not included in this analysis. The cause of these incidents was attributed to factors stemming from crash cases and road segment data. The results indicate that curves accounted for 1.23% of the crashes, while clear slopes and center open areas contributed to 1.19% and 1.06% of the crashes, respectively.

**Table 2.** Variable Description and Descriptive Statistics.

| Variable | Value | Description | Frequency | Percentage |
|---|---|---|---|---|
| Rear-end crash with truck involvement (mean, S.D.) | | | (0.291, 2.071) | |
| Length (mean, S.D.) | | | (3.082, 5.023) | |
| AADT (mean, S.D.) | | | (14,473.9, 24,292.1) | |
| Percent Truck (mean, S.D.) | | | (16.328, 11.771) | |
| Number of lanes per direction | 0 | 2 lanes road | 9707 | 57.32% |
| | 1 | 4 lanes road | 5459 | 32.23% |
| | 2 | 6 lanes road | 1015 | 5.99% |
| | 3 | 6 lanes and more | 755 | 4.46% |
| Speed Limit | 0 | 80 Km/h and less | 2497 | 14.74% |
| | 1 | 90 Km/h | 4794 | 28.31% |
| | 2 | >90 Km/h | 9645 | 56.95% |

**Table 2.** *Cont.*

| Variable | Value | Description | Frequency | Percentage |
|---|---|---|---|---|
| Pavement | 0 | Asphalt concrete | 15,258 | 90.09% |
| | 1 | Concrete | 1678 | 9.91% |
| Lane width | 0 | ≤3 m | 1025 | 6.05% |
| | 1 | 3.1–3.5 m | 15,638 | 92.34% |
| | 2 | >3.5 m | 273 | 1.61% |
| Footpath | 0 | No | 16,131 | 95.25% |
| | 1 | Yes | 805 | 4.75% |
| Shoulder width | 0 | ≤1 m | 6364 | 37.58% |
| | 1 | 1.1–2 m | 3533 | 20.86% |
| | 2 | 2.1–3 m | 6604 | 38.99% |
| | 3 | >3 m | 435 | 2.57% |
| Right-of-Way | 0 | ≤40 m | 11,131 | 65.72% |
| | 1 | 40.1–60 m | 3412 | 20.15% |
| | 2 | 60.1–80 m | 1803 | 10.65% |
| | 3 | >80 m | 590 | 3.48% |
| Median | 0 | Yes | 5608 | 33.11% |
| | 1 | No | 11,328 | 66.89% |
| Median width | 0 | No median | 11,328 | 66.89% |
| | 1 | ≤2 m | 987 | 5.83% |
| | 2 | 2.1–4 m | 664 | 3.92% |
| | 3 | 4.1–6 m | 2438 | 14.40% |
| | 4 | >6 m | 1519 | 8.97% |
| Curve [a] | 0 | Straight | 16,728 | 98.77% |
| | 1 | Curve (R > 100) | 208 | 1.23% |
| Slope [a] | 0 | Normal | 16,734 | 98.81% |
| | 1 | Slope (>3% grade) | 202 | 1.19% |
| Median opening [a] | 0 | Non median opening | 16,757 | 98.94% |
| | 1 | Median Opening with auxiliary lane | 179 | 1.06% |

Note: 0 value is the referent category for the statistical models. [a] counted only the road segment that had a crash at least 1 time.

### 3.2. Model Development

The count model commences by examining the distribution of truck-involved rear-end crashes (TIRC), wherein a significant portion of the count distribution follows a Poisson distribution pattern. This approach was elaborated in the work of Wang, et al. [39]. The probability of road segment *i* having a number of crashes occurring, denoted as $y_i$, can be computed using the following expression:

$$P(y_i) = \frac{\exp(-\lambda_i)\lambda_i^{y_i}}{y_i!} \tag{1}$$

where $P(y_i)$ is the probability of the number of TIRC $y_i$ occurring on road segment *i* and $\lambda_i$ is the Poisson parameter; POI was obtained for each road segment. $E[y_i]$ is the expected number of TIRCs that occur on each road, where $E[y_i]$ is the predicted number of events that occur due to an explanatory variable, for example, traffic surface, curve, or straight line, etc. The relationship between the explanatory variable and the Poisson parameter is in the form of a logarithmic model,

$$\lambda_i = \exp(\beta X_i) \tag{2}$$

where $X_i$ the explanatory vector and $\beta$ is the parameter interpolation vector. The number of events or the number of TIRCs was predicted using $E[y_i] = \lambda_i = \exp(\beta X_i)$, and the model was predicted via the maximum likelihood methods.

$$L(\beta) = \prod_i \frac{\mathrm{EXP}[-\mathrm{EXP}(\beta X_i)][\mathrm{EXP}(\beta X_i)]^{y_i}}{y_i!} \tag{3}$$

The Log of the likelihood function is easier to handle and more suitable for estimation and can be calculated as follows:

$$L(\beta) = \sum_{i=1}^{n} [-\exp(\beta X_i) + y_i \beta X_i - LN(y_i!)] \tag{4}$$

The negative Binomial Regression Model (NBR) is used when the results of Poisson's deviations are not appropriate, i.e., the mean of the estimates is not equal to the variance. If the expected value is greater than the variance, it is called under-dispersed ($E[y_i] > VAR[y_i]$), or, most often, over-dispersed ($E[y_i] < VAR[y_i]$). The negative Binomial Regression Model is modified from Equation (2) [40,41]

$$\lambda_i = \text{EXP}(\beta X_i + \varepsilon_i) \tag{5}$$

where $\text{EXP}(\varepsilon_i)$ is Gamma-distribution with a mean of 1 and variance $\alpha$, which is added so that the variance can vary from the mean.

$$VAR[y_i] = E[y_i][1 + \alpha E[y_i]] = E[y_i] + \alpha E[y_i]^2 \tag{6}$$

The probability equations of the Poisson model were set to $\alpha = 0$, which means that the choice between two models (Poisson and negative binomial) depends on the value of $\alpha$, which is most often over-dispersed. The probability of TIRC occurring on a road segment that is considered to have a negative binomial distribution is calculated as follows:

$$P(y_i) = \frac{\Gamma\left(\left(\frac{1}{\alpha}\right) + y_i\right)}{\Gamma\left(\frac{1}{\alpha}\right) y_i!} \left(\frac{\frac{1}{\alpha}}{\left(\frac{1}{\alpha}\right) + \lambda_i}\right)^{\frac{1}{\alpha}} \left(\frac{\lambda_i}{\left(\frac{1}{\alpha}\right) + \lambda_i}\right)^{y_i} \tag{7}$$

where $\Gamma(.)$ is the gamma function. For parameter estimation, we could obtain:

$$P(\lambda_i) = \prod_i \frac{\Gamma\left(\left(\frac{1}{\alpha}\right) + y_i\right)}{\Gamma\left(\frac{1}{\alpha}\right) y_i!} \left(\frac{\frac{1}{\alpha}}{\left(\frac{1}{\alpha}\right) + \lambda_i}\right)^{\frac{1}{\alpha}} \left(\frac{\lambda_i}{\left(\frac{1}{\alpha}\right) + \lambda_i}\right)^{y_i} \tag{8}$$

When forecasting the number of crashes that occur on each road each year, there may be some roads where, throughout the information, no crashes are recorded. This can be separated into two different event characteristics. It contains normal numbers (normal count), and the nature of the count is zero (zero count). The NBR model is not covered for splitting the analysis into two parts. Therefore, a suitable model for dual-state is the zero-inflated model. When built on top of the negative binomial model, it is called the zero-inflated model (ZINB) [42]

The ZINB model is similar to the occurrence equation $Y = (y_1, y_2 \ldots, y_n)$, which is independent.

$$\begin{aligned} y_i &= 0 \text{ with probability } p_i + (1 + p_i)\left(\frac{\frac{1}{\alpha}}{\left(\frac{1}{\alpha}\right) + \lambda_i}\right)^{\frac{1}{\alpha}} \\ y_i &= y \text{ with probability } (1 + p_i)\left(\frac{\Gamma\left(\left(\frac{1}{\alpha}\right) + y\right) \mu_i^{\frac{1}{\alpha}} (1 - \mu_i)^y}{\Gamma\left(\frac{1}{\alpha}\right) y_i!}\right), \quad y = 1, 2, 3, \ldots \end{aligned} \tag{9}$$

where $\mu_i = \left(\frac{1}{\alpha}\right)\left[\frac{1}{\alpha} + \lambda_i\right]$ is the maximum likelihood method used to re-estimate the parameters in the ZINB model. For parameter estimation using the maximum likelihood method, confidence was determined via the likelihood ratio test.

In some cases, it may be reasonable to assume that there is a correlation between road physical characteristics and land use. This relationship occurs due to spatial considerations, such as crashes that occur in the same area. It should be set to have an unobservable

influence. In this study, the division of roads into the Department of Highway branch (DOHB)-supervised areas is proposed. In such a relationship, model applications can generate random effects when an unobservable influence is considered an indicator variable.

The equation for the random influence of counts, improved from Equation (5), resulted in the following:

$$LN(\lambda_i) = \beta X_i + \varepsilon_i + \eta_i \text{ or, } \lambda_i = \text{EXP}(\beta X_i + \varepsilon_i)\text{EXP}(\eta_i) \tag{10}$$

where $\lambda_i$ is the expected number of events that occur for road segment *i*, which falls into group *j* (the DOHB regulatory region expected to exert unobserved heterogeneity). $X_i$ is a vector of explanatory variables, $\beta$ is the parameter prediction vector and $\eta_i$ is the random influence for dataset *j*. Spatial correlation was calculated for the spatial variation proportion out of the total variation [13] as follows:

$$Spatial\ correlation = \frac{\sigma_\eta^2}{\sigma_\eta^2 + \sigma_e^2} \tag{11}$$

where $\sigma_\eta^2$ is the predicted variance $\eta_j$ or the intra-space variance of the same DOHB. $\sigma_e^2$ is the variance obtained from the fixed effect parameter estimates or the variance that occurs between DOHB areas of responsibility. A general model can be obtained assuming that $\eta_j$ is randomly distributed through each group, i.e., $\text{EXP}(\eta_j)$ is gamma-distributed with an average of one and has a variance of $\alpha$.

Crash modification factors (CMF) were considered the same as the odds ratio (OR), which was calculated to evaluate a change in the relative risk of TIRC due to a change in an unmatched explanatory variable, which, in this study, was a difference in the visual characteristics of the road, calculated from a dummy variable [43].

$$OR(x_i) = \text{EXP}(\beta_i) \tag{12}$$

The data analysis of this study was performed using the R programming language along with the core software package 'stats' version 3.6.2. In addition to this, specific packages such as 'readr' and 'dplyr' were employed for data analysis and preparation. Moreover, the 'pscl::zeroinfl' package was utilized to analyze the ZINB model [44], which stands for zero-inflated negative binomial regression. Furthermore, the SZINB model was analyzed using 'glmmTMB' (Generalized Linear Mixed Models with Template Model Builder) [45].

## 4. Results

### 4.1. Model Statistics

The statistical results of the model are presented in Table 3 and initially developed from the Poison Regression Model (POI). The McFadden $\rho^2$ value stands notably high at 0.04010. However, the predicted dispersion ratio was 4.676, indicating the importance of having a *p*-value < 0.000. As a result, it became crucial to develop the Negative Binomial Regression Model (NBR), and the mean zero probability forecast was 0.910. Many road segments exhibited zero crashes, necessitating the development of a zero-inflated Negative Binomial Regression (ZINB) Model. To determine the most suitable model, the Akaike Information Criterion (AIC) was considered, which revealed that the AIC of NBR (12,946.76) was higher than that of ZINB (12,416.14). This suggests that the model with zero state separation is more appropriate. Additionally, the dispersion ratio serves as an indicator of the suitability of the negative binomial model. Discrepancies between the Department of Highways Branch (DOHB)-controlled areas were acknowledged, and a spatial correlation analysis indicated an intra-area variance of 0.396 (39.6%) [46]. Consequently, it was deemed appropriate to incorporate the random effect into the ZINB model, leading to the development of the Spatial Zero-Inflated Negative Binomial Regression (SZINB). The AIC value for SZINB was calculated to be 12,198.59, which is lower than that of ZINB,

suggesting that SZINB is more suitable. Thus, parameter interpretations should be derived from the SZINB model based on the research by Raihan, et al. [23], considering its superior suitability compared to ZINB.

**Table 3.** Models' Statistics.

| Model Statistics | POI | NBR | ZINB | SZINB |
|:---|:---:|:---:|:---:|:---:|
| Log−likelihood intercept-only model | −17,834.9 | −7348.39 | −7348.39 | −6984.54 |
| Log−likelihood convergence model | −10,682.3 | −6448.38 | −6159.07 | −6049.3 |
| McFadden $\rho^2$ | 0.4010 | 0.1225 | 0.1618 | 0.1339 |
| The Akaike Information Criterion (AIC) | 21,412.55 | 12,946.76 | 12,416.14 | 12,198.59 |
| Mean zero probability | 0.910 | | | |
| Dispersion ratio | 4.676 *** | | | |
| Over-dispersion | | | 0.841 | |
| Spatial correlation | | | | 0.396 |

Note: *** *p*-value < 0.000. POI denotes Poisson Regression. NB denotes Negative Binomial Regression. ZINB denotes Zero-inflated Negative Binomial Regression. SZINB denotes Spatial Zero-Inflated Negative Binomial Regression. The Akaike Information Criterion (AIC) is calculated as follows: AIC = −2 × log-likelihood + 2 × K, where k is the number of estimated parameters in the model. McFadden $\rho^2$ is calculated as: $\rho^2 = 1 - LL(m1)/LL(m0)$.

### 4.2. Correlations and Parameter Estimations

Figure 2 illustrates the correlation among all variables pertaining to rear-end crashes involving trucks. Notably, the TIRC exhibited a remarkable influence. The variables that showed a positive impact on these crashes included colliding with curved roads, areas with slope road alignment, and the number of traffic lanes. Moreover, certain road environmental characteristics, such as the length of the road section and traffic volume, also contributed significantly to the occurrence of these crashes. Within this array of variables, the presence of a center island and, notably, the width of the center island emerged as possessing the most substantial correlations with rear-end crashes. Consequently, in the course of our research model development, we exclusively prioritized the consideration of the variable linked to the width of the center island, owing to its pronounced and statistically significant association with the incidence of rear-end collisions involving trucks.

Table 4 presents a conditional model of the SIZNB (Spatial Interdependency Zero-Inflated Negative Binomial) model, illustrating the correlation between road geometry design and the frequency of TIRC with one or more occurrences. The analysis revealed that several factors exerted an impact on the frequency of crashes. Firstly, the number of lanes played a crucial role, with 6-lane roads exhibiting a higher probability of TIRC (odds ratio or crash modification factor; CMF = 1.5914) when compared to roads with 1 lane per direction used as a reference. Conversely, roads with 4 lanes and 2 lanes did not show significant effects. Moreover, it was observed that higher speeds (>90 Km/h) led to an increased likelihood of TIRC (CMF = 1.7229) compared to speeds below 90 Km/h. Additionally, concrete road surfaces demonstrated lower TIRC rates than asphalt concrete road surfaces (CMF = 0.6658). Various other variables were found to contribute to elevated TIRC levels. These included the presence of footpaths on roads (CMF = 1.4393) compared to roads without footpaths, increased shoulder width compared to roads without shoulders (CMF = 1.4393–1.6240), and wider right-of-way widths (CMF = 1.3967–1.6158). Furthermore, when considering variables associated with the crash's physical characteristics and their match with the road's attributes, it was evident that certain conditions significantly elevated the probability of TIRC occurrence. Such conditions encompassed curved road sections (CMF = 5.0369), sloped road sections (CMF = 2.1089), and central island open areas (CMF = 4.9738). These factors were identified as having a substantial impact on the frequency of TIRC in this study.

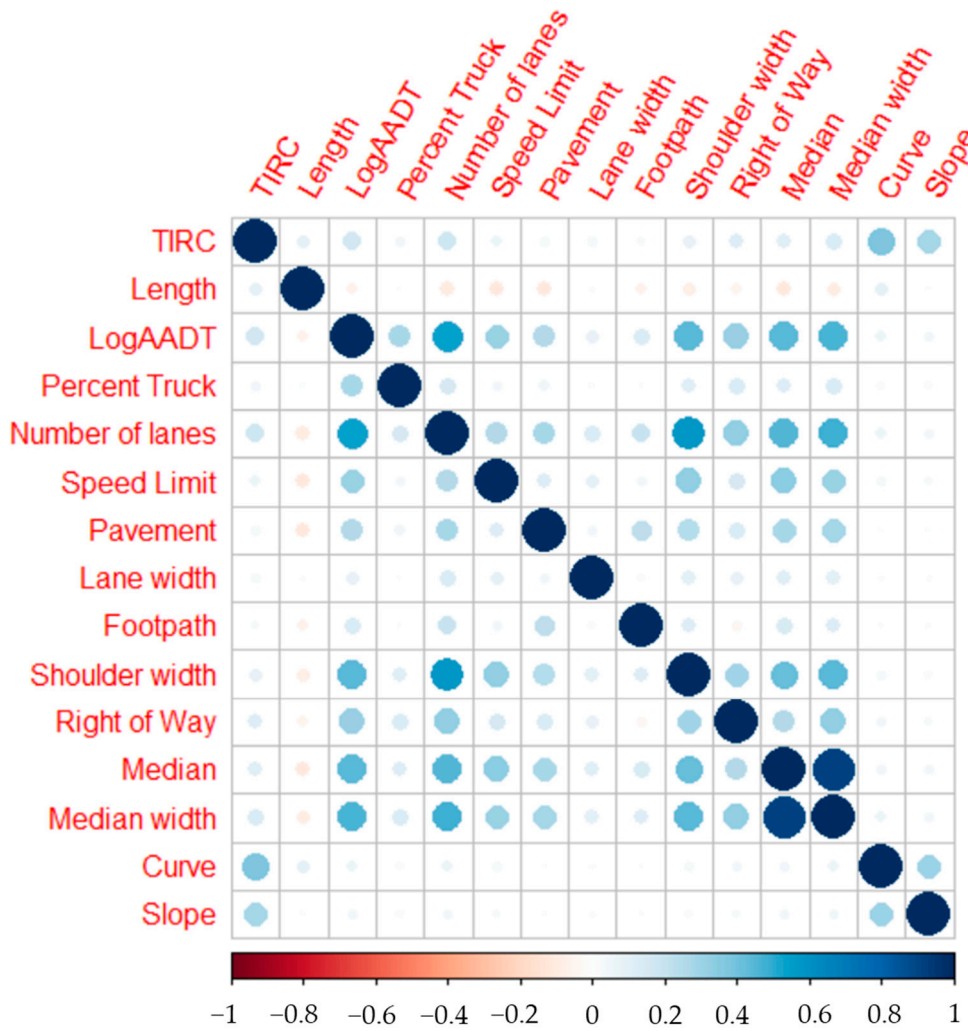

**Figure 2.** Correlation between explanatory variables in truck-involved rear-end crashes (TIRC). The depiction of the factors is in reference to Table 2.

**Table 4.** SZINB results.

|  | Estimate | S.D. | t-Stat | *p*-Value | Sig. | CMF |
|---|---|---|---|---|---|---|
| Random Effect (intercept) | 5.970 | 2.443 | 2.443 | 0.0201 | ** |  |
| Conditional model: |  |  |  |  |  |  |
| (Intercept) | −1.363 | 0.313 | −4.361 | <0.000 | *** |  |
| 2 lanes road | 0.114 | 0.132 | 0.865 | 0.387 |  |  |
| 4 lanes road | 0.240 | 0.183 | 1.313 | 0.189 |  |  |
| 6 lanes road | 0.465 | 0.178 | 2.611 | 0.009 | ** | 1.5914 |
| 90 Km/h | −0.074 | 0.214 | −0.345 | 0.730 |  |  |
| >90 Km/h | 0.544 | 0.134 | 4.049 | <0.000 | *** | 1.7229 |
| Concrete Pavement | −0.407 | 0.119 | −3.423 | 0.001 | ** | 0.6658 |
| Lane width [3.1–3.5 m] | −0.219 | 0.271 | −0.808 | 0.419 |  |  |
| Lane width [>3.5 m] | −0.578 | 0.385 | −1.499 | 0.134 |  |  |
| Footpath | 0.364 | 0.164 | 2.226 | 0.026 | ** | 1.4393 |
| Shoulder width [1.1–2 m] | 0.359 | 0.154 | 2.324 | 0.020 | ** | 1.4318 |
| Shoulder width [2.1–3 m] | 0.485 | 0.152 | 3.180 | 0.001 | ** | 1.6240 |
| Shoulder width [>3 m] | 0.134 | 0.234 | 0.574 | 0.566 |  |  |
| Right-of-way [40.1–60 m] | 0.447 | 0.111 | 4.017 | <0.000 | *** | 1.5643 |
| Right-of-way [60.1–80 m] | 0.334 | 0.133 | 2.518 | 0.012 | ** | 1.3967 |

**Table 4.** *Cont.*

|  | Estimate | S.D. | t-Stat | *p*-Value | Sig. | CMF |
|---|---|---|---|---|---|---|
| Right-of-way [>80 m] | 0.480 | 0.207 | 2.319 | 0.020 | ** | 1.6158 |
| Median width [≤2 m] | 0.119 | 1.509 | 0.079 | 0.937 | | |
| Median width [2.1–4 m] | 0.278 | 1.507 | 0.185 | 0.853 | | |
| Median width [4.1–6 m] | 0.412 | 1.490 | 0.276 | 0.782 | | |
| Median width [>6 m] | 0.564 | 1.495 | 0.377 | 0.706 | | |
| Curve | 1.617 | 0.120 | 13.502 | <0.000 | *** | 5.0369 |
| Slope | 0.746 | 0.192 | 3.881 | <0.000 | *** | 2.1089 |
| Median opening | 1.604 | 0.161 | 9.948 | <0.000 | *** | 4.9738 |

Note: Sig. denotes significant level. *** *p*-value < 0.001, ** *p*-value < 0.05. CMF denotes crash modification factor (odd ratio).

## 5. Discussion

For the random effect of the intercept, the estimated value was found to be 5.970 with a standard deviation (S.D.) of 2.443 (*p*-value < 0.05), signifying a statistically significant difference in truck-involved rear-end crashes (TIRC) occurrences in the control area of the Department of Highways branches. This disparity is likely attributable to variations in land use across different provinces. For instance, certain provinces may prioritize the development of industrial estates or establish connections with ports, leading to a higher volume of truck traffic and subsequently increasing the likelihood of TIRC incidents. On the other hand, provinces focusing on tourism or agriculture can experience fewer TIRC cases due to reduced truck activity in such regions. The following discussion is considered from the context of highways in Thailand, which may have different characteristics from other countries.

An assessment of the intercept parameter yielded a value of −1.363 (<0.000), indicating that the TIRC for most road segments was 0, corresponding to a mean probability of 0. Regarding the variables related to the number of lanes, the reference variable was set as one lane per direction. The parameter prediction results revealed that there were only six traffic lanes per direction, which could result in a higher likelihood of TIRC occurrences. This is because roads with 2 and 4 lanes per direction showed no significant impact on TIRC, possibly due to the minimal difference in the number of TIRCs. The reason behind the higher TIRC likelihood for roads with six lanes per direction is that these roads are designed to facilitate maximum vehicular movement, especially in Thailand, where they are frequently constructed to accommodate large vehicles. As the number of cars increases on such roads, the probability of collisions with the rear of trucks is elevated. This observation aligns with the findings of Lao, et al. [19], who also reported that an increase in the annual average daily traffic (AADT) or traffic volume leads to a higher frequency of rear-end collisions.

For speed variables, the reference value used was <90 km/h, which roughly corresponds to a range of approximately 60–80 km/h. The parameter evaluation revealed that a speed of 90 km/h. was not statistically significant, while speeds exceeding 90 km/h. showed a significant positive correlation with an increased likelihood of TIRC occurring and a potential frequency surge of about 72.3%. This finding is highly plausible and aligns with numerous previous studies, including those conducted by Peng, et al. [47], Wang, et al. [48]. The rationale behind this outcome is rooted in the fact that rear-end collisions tend to happen more frequently in areas characterized by high-speed roads, such as highways connecting cities or expressways designed for swift travel. These roads are particularly susceptible to higher rates of rear-end collisions due to the challenges of braking in time for vehicles moving at such speeds.

Concrete road surfaces exhibit a lower susceptibility to Tire-Induced Roadway Cracking (TIRC) compared to asphalt concrete road surfaces. This disparity can be attributed, in all likelihood, to the higher frictional properties inherent in concrete road surfaces. This phenomenon aligns with the findings of Iranitalab, et al. [29], who demonstrated that cargo

tank truck crashes leading to rollovers and the release of hazardous materials are less probable on concrete roads in comparison to asphalt concrete roads.

The lane width variable was assigned a value of less than 3.1 m as a reference in this study. However, the results obtained from this value were deemed insignificant in the model. This finding contrasts with prior research conducted by Saeed, et al. [41], Moomen, et al. [49], which indicated that increasing the width of traffic lanes can lead to a reduction in the frequency of collisions.

Despite this inconsistency, the reason behind the insignificant results could be adequately explained, regardless of whether the traffic lane is wider or narrower. The key factor appears to be the driving behavior of most trucks, as they predominantly operate on highway roads with larger lane widths. Consequently, these roads encompass both high-collision areas and regions with lower collision rates. This blend of road types dilutes the impact of lane width on collision frequency, rendering it an insignificant factor in the model. A similar investigation by Moomen, et al. [50] produced similar findings, where lane width was not significantly associated with the frequency of truck crashes on downgraded roads.

In the context of the variable concerning the presence of a footpath, it was deemed to be significant, leading to an approximately 43.9% higher incidence of TIRC. This finding contrasts with a study conducted by Champahom, et al. [18], where they observed that the presence of a footpath did not show significance in relation to the frequency of rear-end collisions for all types of vehicles. However, the underlying cause of this discrepancy is likely attributable to roads with footpaths situated in urban areas. In such areas, traffic tends to maintain a relatively low headway, and trucks with inadequate front spacing exhibit specific behavioral patterns. Consequently, the likelihood of rear-end collisions increases considerably. This aligns with the conclusions of Lao, et al. [19], who similarly reported a higher probability of rear-end collisions in urban areas, specifically on rural roads.

The lane width factor exhibits multiple values, with the reference value corresponding to a shoulder width of $\leq 1$ m, which is deemed excessively narrow. These findings indicate that the critical variables are a shoulder width of 1.1–2 m and 2.1–3 m, which consequently raise the likelihood of TIRC occurrence. Interestingly, these results contradict the study conducted by Lao, et al. [19], which suggested that an increased road shoulder width is inversely associated with rear-end crash risk. Nevertheless, plausible explanations can be provided. A significant portion of roads with broad shoulders falls within the category of intercity roads, which facilitates high-speed travel and often accommodates parked trucks on the roadside. Consequently, these areas are more susceptible to TIRC incidents. This assertion is supported by a study conducted by Champahom, et al. [18], which revealed that collisions predominantly transpire on highways or roads that prioritize mobility. The variables of shoulder width and right-of-way (ROW) share a high correlation, as depicted in Figure 2, which also enhances the likelihood of TIRC occurring. The underlying reason remains consistent for roads with numerous boundaries, frequently serving as conduits between major cities, which tend to have a higher volume of traffic channels and are commonly used at high speeds, thereby elevating the possibility of TIRC frequencies. This correlation is further substantiated by the evidence presented in Figure 2.

In this model, the variable pertaining to the width of the central island was deemed insignificant. Conversely, the number of traffic lanes, the ROW, and relatively high shoulder width were identified as variables with positive implications for TIRC. However, despite their positive impact, these variables were also found to be statistically insignificant in the model. To elaborate, the width of the middle island does not directly influence parking on the shoulder or the flow of traffic speed. Consequently, it does not exert a significant effect on TIRC. This observation aligns with the findings of Champahom, et al. [18], who similarly concluded that the width of the central island does not play a significant role in the frequency of rear-end collisions.

The curved road area, in comparison to the reference variable representing the straight road area, is characterized by a distinct feature. This distinctiveness is evident in its high parameter value, which indicates a significant likelihood of encountering a TIRC. This

finding is consistent with various previous studies, including the research conducted by Wang, et al. [25], who elaborated on the reason behind this correlation, explaining that the curved road area often serves as a location where trucks slow down or come to a halt before entering the curve. Consequently, if the vehicle following behind fails to exercise caution or is unable to brake promptly, there is an increased risk of a rear-end collision. Furthermore, Ding, et al. [51] emphasized the importance of rear-end crashes on curved roads, with a particular focus on factors such as visual perception and traffic flow uncertainty. Their study further corroborated the association between curved road areas and heightened crash probabilities. Notably, some curved road areas may be inadequately equipped with warning signs, as they may not meet the standard requirements. For instance, the positioning of warning signs before the curve might be inappropriate, or the clarity of the signs themselves may be lacking. Such deficiencies could potentially contribute to an increased risk of crashes in these areas.

For roads exhibiting slope characteristics, a noteworthy increase in the likelihood of TIRCs was observed, aligning with the findings of Wen, et al. [2]. Their research emphasized that steep slopes significantly elevate the risk of multi-vehicle collisions involving trucks. This heightened risk can be attributed to the additional weight burden on trucks when navigating such inclines, necessitating the extensive use of heavy brakes, especially during descents, where repeated braking may result in rear-end collisions between trucks and other vehicles ahead, as the latter attempt to maintain their speed. This scenario is particularly relevant in the context of Thailand, where vital routes frequently witness heavy truck traffic. These routes serve as crucial links between the eastern industrial zone and the densely populated northern region, rendering them high-traffic areas for trucks and other vehicles.

The initial entry point to the central island is regarded as one of the slower sections or frequently necessitates abrupt braking. As a consequence, there is a significantly elevated likelihood of rear-end collisions, as evidenced by the CMF values indicating a substantially higher probability of these occurrences compared to areas without a central island. This phenomenon is corroborated by a study conducted by Champahom, et al. [37], which identified the central island launch area as a hotspot for rear-end crashes. In such incidents, it is common for the truck to collide with the vehicle in front.

## 6. Conclusions and Implementations

This study aimed to conduct an analysis of the factors pertaining to road characteristics that significantly influence the frequency of rear-end collisions involving trucks. The dataset utilized for this investigation comprises crash data recorded on highways during the period from 2011 to 2018.

In terms of methodological contribution, after careful consideration of the Akaike Information Criterion, we determined that the most efficient simulation method to employ is the Spatial Zero-Inflated Negative Binomial Regression (SZINB). This method is particularly valuable due to its incorporation of spatial correlation, which is a crucial aspect of our analysis. Overlooking this factor could result in biased parameter estimation, thereby impacting the accuracy of our statistical analysis in crash frequency studies.

The findings of this study suggest that there are discernible disparities in truck-involved rear-end crashes (TIRC) when occurring on highway roads. The estimation of parameters within the SZINB model identified several key factors contributing to a higher probability of TIRC frequencies. These factors include road curvature compared to straight stretches, road areas that serve as the opening points of central islands, road segments with wide middle grips and ample right-of-way, roads featuring six traffic lanes per direction (compared to roads with just one lane per direction), roads equipped with footpaths, and roads with a speed limit exceeding 90 km/h.

This study's contribution lies in the form of policy or measure recommendations. These recommendations are aimed at addressing the issue of TIRC reduction by considering the visual design of the road. The suggestions presented in this study were developed based on key findings, particularly focusing on the high value of the crash modification factor (CMF).

To begin with, this study suggests implementing measures to enhance the awareness of drivers as they approach proper curves. This can be achieved by inspecting and installing signs in these areas. Furthermore, conducting a road safety audit on curves with a history of frequent crashes is proposed as an initial step. Regarding the central island opening area, this study recommends making drivers aware of the opening point in the middle of the island. This can be achieved by considering the appropriate position to install warning signs. To address TIRC on steep slopes, this study proposes requiring trucks to use lower gears. This could reduce the need for excessive braking. Additionally, installing signs or checkpoints before descending hills can remind drivers to use lower gears. The study also identified areas where speed exceeds 90 km/h. Although trucks typically do not travel at such high speeds, there is a risk of other vehicles colliding with the rear of the truck. Therefore, this study suggests considering road sections with high-speed usage and a high frequency of crashes. In such areas, the installation of traffic equipment, such as two dots to indicate proper spacing or rear-end collision warning signs, is recommended to prevent rear-end collisions.

In summary, this study recommends various measures to address TIRC reduction, including enhancing driver awareness on curves, installing warning signs at critical points, promoting the use of lower gears on steep slopes, and implementing traffic equipment in areas with high-speed usage and crash rates. These policy and measure recommendations aim to improve overall road safety and reduce traffic-related incidents and crashes.

In addition to its relevance in the context of Thailand, the findings of this study hold promise for countries within the same region that share similarities in culture, economic conditions, and road infrastructure challenges. Several nations in Southeast Asia, for instance, exhibit comparable characteristics in terms of road systems and traffic dynamics. These countries may draw valuable insights from this study's recommendations to address TIRCs and enhance overall road safety. Countries facing challenges in relation to road curvature and the presence of central islands as access points could consider implementing awareness campaigns and strategic signage placements similar to the suggestions put forth in this study. Additionally, the promotion of lower gear usage on steep slopes is a universally applicable safety measure that can benefit regions with hilly terrains. Furthermore, nations grappling with high-speed usage on certain road sections and an increased frequency of rear-end collisions can explore the installation of traffic equipment, such as indicators for safe following distances or rear-end collision warning signs. These measures align with the broader goal of minimizing TIRCs and improving road safety, making them pertinent for countries in the region facing similar challenges. By sharing and adapting the insights and recommendations gleaned from this study, neighboring countries can collectively work toward reducing traffic-related incidents and enhancing road safety, ultimately fostering safer and more efficient transportation networks across the region.

This investigation's findings have unveiled regional variations in rear-end truck collisions. However, it is essential to acknowledge the extensive geographical scope of this study. Consequently, future research should prioritize specific locations, such as Bangkok, Pathum Thani, and Chonburi, where urgent corrective measures are needed. To enhance the precision of recommendations, supplementing this study with on-site surveys is advisable. Additionally, in the academic context, opportunities exist to deepen our understanding by investigating significant factors. For instance, further research could delve into the impact of curves on the heightened occurrence of TIRC. Investigating these curves in the context of Naturalistic Driving among truck drivers, considering various radii and contextual settings, could provide valuable insights into driver responses.

**Author Contributions:** Conceptualization, T.C.; methodology, T.C.; software, T.C. and C.S.; validation, C.S.; formal analysis, T.C.; investigation, S.J. and R.K.; data curation, T.C. and V.R.; writing—original draft preparation, T.C.; writing—review and editing, T.C. and C.S.; visualization, T.C.; supervision, V.R. and S.J.; project administration, S.J. and R.K.; funding acquisition, S.J. All authors have read and agreed to the published version of the manuscript.

**Funding:** This work was supported by Suranaree University of Technology (SUT), Thailand Science Research and Innovation (TSRI), and National Science, Research and Innovation Fund (NRSF) (NRIIS number 179277).

**Institutional Review Board Statement:** This research was approved by the Ethics Committee for Research Involving Human Subjects, Suranaree University of Technology (EC-65-99).

**Informed Consent Statement:** Not applicable.

**Data Availability Statement:** Data are available on request due to restrictions, e.g., privacy or ethical.

**Acknowledgments:** The authors would like to thank the Department of Highways for supporting the road traffic crash and road geometry data.

**Conflicts of Interest:** The authors declare no conflict of interest.

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
