# Peer review of "Analysis of the Effects of Highway Geometric Design Features on the Frequency of Truck-Involved Rear-End Crashes Using the Random Effect Zero-Inflated Negative Binomial Regression Model"

_safety, 2023_

Round 1

Reviewer 1 Report

Comments and Suggestions for Authors

This paper describes the modelling of truck rear-end crashes. The results show the geometric design features that are related to these type of crashes.

 Abstract, lines 23/24

The use of ‘and so on’ in an abstract is not very informative. Please skip this expression.

Page 2, line 63

Crashes are always the outcome of a combination of variables regarding three factors: driver(s), vehicle(s) and the road environment.

Page 10, lines 335/336

Is the truck load ratio part of your analysis? This variable is not visible in Table 2.

Page 11, Figure 3

The variable Footpath is written as Footbath…

Page 11, lines 343 and 364

The headings of Figure 3 and Table 4 should be more descriptive.

Reviewer 2 Report

Comments and Suggestions for Authors

Dear authors; many thanks for the oportunity to revise this work. The paper is well structured and it is the written is clear and I am grateful to them for this.

Reviewer 3 Report

Comments and Suggestions for Authors

General

The topic of the study “Analysis of The Effects of Highway Geometric Design Feature on Frequency of Truck Involved Rear-End Crashes Using Random Effect Zero-Inflated Negative Binomial Regression Model” is of relevance to the Safety journal readership.

The objective of this study is to create a model that analyzes the factors influencing the frequency of rear-end crashes involving trucks. The authors propose a model using Spatial Zero-inflated Negative Binomial Regression. The methodology is well described and well demonstrated.

The paper is well written and informative.

Major Comments

The study is based on data from Thailand. The choice of Thailand should be explained and justified. Also, the findings and conclusions are limited to the context of Thailand’s highways – please have them extended to t a wider context.

The authors do not provide specific research questions/ hypotheses.

The limitations of this study (briefly mentioned in the last paragraph) should be further discussed.

Minor Comments

Figure 1 and Table 1 provide the same information about crash severity and collision type; one of them can be removed (possibly the figure).

Figure 2 – it is not clear what the region colors represent. Either provide an explanation or remove the colors.

Figure 2 – the figure is not clear - please increase the figure’s size and improve its resolution.

Table 2- indicate what the ‘a’ sub-comment stands for.

Figure 3 – the title ‘correlations’ provides very little explanation of the figure.

Reviewer 4 Report

Comments and Suggestions for Authors

The topic of the paper is relevant. The statistical methods used are highly sophisticated. However, I find that it has a basic problem.

“In summary, the study recommends various measures to address TIRC reduction, including enhancing driver awareness on curves, installing warning signs at critical points, promoting the use of lower gears on steep slopes, and implementing traffic equipment in areas with high-speed usage and accident rates. These policy and measure recommendations aim to improve overall road safety and reduce traffic-related incidents and crashes. To address TIRC on steep slopes, the study proposes requiring trucks to use lower gears. This would reduce the need for excessive braking.”

These kind of findings are obvious, the paper itself refers to relevant literature studies. For me it is not clear, what is the added value of the paper.

Comments on the Quality of English Language

Please explain the content of figure 3 in more detail.

Round 2

Reviewer 1 Report

Comments and Suggestions for Authors

The new version is acceptable for publication.

Reviewer 2 Report

Comments and Suggestions for Authors

The authors have improved this version.

It is a minor error in text  in Eq. 9=

𝑝𝑟𝑜𝑎𝑏𝑖𝑙𝑖𝑡𝑦 instead p𝑟𝑜b𝑎𝑏𝑖𝑙𝑖𝑡𝑦

With this correction the article is candidate for publication.

Reviewer 3 Report

Comments and Suggestions for Authors

The authors have well addressed my comments/ I have no further comments.

Reviewer 4 Report

Comments and Suggestions for Authors

My comments have been properly addressed.

Round 3

Reviewer 2 Report

Comments and Suggestions for Authors

The authors have been addressed my comments properly. Thank you.